# Re-evaluating Minimum Bayes Risk Decoding for Automated Speech Recognition Tasks

**Yuu Jinnai**                                                                                      *jinnai_yu@cyberagent.co.jp*
*CyberAgent*

**Reviewed on OpenReview:** *https://openreview.net/forum?id=I6iLWhRIsf*

## Abstract

While sample-based Minimum Bayes Risk (MBR) decoding has shown to outperform beam search in many text-to-text generation tasks with modern LLMs, beam search remains the dominant approach for Automatic Speech Recognition (ASR) and Speech Translation (ST). To date, the efficacy of MBR decoding within modern speech systems lacks comprehensive evaluation. Given that MBR decoding is effective in text-to-text generation tasks, it is reasonable to expect it to also be effective for speech-to-text tasks. In this paper, we evaluate MBR decoding for ASR and ST tasks on English and Japanese using Whisper and its derivative models. We observe that the accuracy of MBR decoding outperforms that of beam search in most of the experimental settings we have evaluated. The results show that MBR decoding is a promising method for ASR and ST tasks that require high accuracy. The code is available at `https://github.com/CyberAgentAILab/mbr-for-asr`.

## 1 Introduction

Automatic Speech Recognition (ASR) is the task of converting spoken language into written text and plays a crucial role in a wide range of applications. Advances in deep learning have significantly improved the accuracy and robustness of ASR systems, enabling their deployment in diverse real-world scenarios (Prabhavalkar et al., 2024).

Decoding algorithms play an important role in determining the final output quality of ASR systems. One of the common approaches, beam search, incrementally explores the most probable hypotheses to approximate the maximum-a-posteriori (MAP) solution. While effective and efficient, beam search is known to suffer from several degeneration issues in text-to-text generation tasks such as machine translation (Holtzman et al., 2020; Eikema & Aziz, 2020). Minimum Bayes risk (MBR) decoding offers a promising alternative by directly optimizing for the expected utility of the output (Goel & Byrne, 2000; Kumar & Byrne, 2004). Rather than selecting the single most probable sequence, MBR considers multiple candidate hypotheses and chooses the one that minimizes the expected loss (or maximizes utility) when compared against other likely outputs (Bickel & Doksum, 2015). This approach has shown remarkable success in text-to-text tasks such as machine translation, summarization, and captioning (Eikema & Aziz, 2022; Suzgun et al., 2023; Jinnai et al., 2024; Wu et al., 2025), consistently outperforming beam search across diverse evaluation metrics.

While MBR decoding has been evaluated for classic ASR systems (e.g., hidden Markov model, Goel & Byrne 2000; Goel et al. 2004), its application to speech-to-text tasks with the modern ASR systems has not been investigated (Prabhavalkar et al., 2024). For example, MBR decoding has been applied to the spoken language translation in the recent IWSLT shared tasks (Ahmad et al., 2024; Abdulmumin et al., 2025), but it is used for the machine translation modules rather than the ASR modules of the cascaded systems (Yan et al., 2024; Ben Kheder et al., 2024; Li et al., 2024; Wang et al., 2025; Romney Robinson et al., 2025).

Given that the method is designed to improve the decoding accuracy of probabilistic models in general (Ichihara et al., 2025a), it is reasonable to expect it to also improve the accuracy of ASR modules. The absence of comprehensive studies on MBR decoding for contemporary ASR systems represents a significant

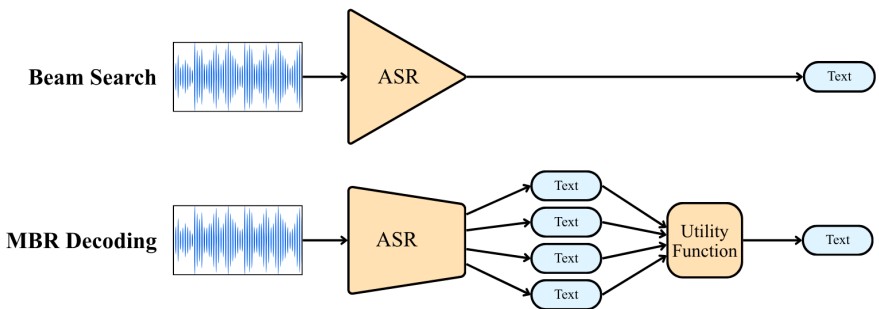

Figure 1: Illustration of the beam search and MBR decoding: multiple hypotheses are sampled from the ASR model, and the hypothesis with the highest expected utility (e.g., BLEU score) compared to the others is selected as the final output.

gap in the literature. Given MBR's empirical successes in text-to-text tasks and theoretical advantages, a systematic evaluation of its potential for speech recognition is valuable.

To this end, we present a comprehensive evaluation of sample-based MBR decoding for both offline ASR and Speech Translation (ST) tasks (Figure 1). Our experiments span multiple languages, with a focus on English and Japanese. We use diverse datasets, multiple models based on Whisper, and with varying levels of synthesized noise added. MBR decoding consistently outperforms beam search across these dimensions, often by substantial margins. Remarkably, these improvements emerge with as few as 4-8 samples, suggesting that MBR can be practically implemented in scenarios where latency requirements are not stringent.

Our findings have significant implications for high-accuracy ASR applications where transcription quality takes precedence over real-time processing. While the computational overhead of MBR makes it less suitable for real-time applications, its consistent accuracy improvements make it an attractive option for offline speech-to-text systems. This work thus reestablishes MBR decoding as a valuable technique in the modern neural ASR toolkit.

## 2 Background

We first formally define the text generation problem and then describe MBR decoding.

### 2.1 Text Generation Problem

Conditional text generation is the problem of generating a sequence of tokens $y \in \mathcal{Y}$ conditioned on an input context $x$, using a probabilistic model $P(y|x)$, where $\mathcal{Y}$ is the set of all possible sequences (Graves, 2012; Sutskever et al., 2014). Formally, we denote $\mathcal{V}$ as a set of tokens (vocabulary). Let bos and eos be special tokens representing the beginning and end of a sequence, respectively. Then, $\mathcal{Y}$ is the set of sequences of tokens from the vocabulary $\mathcal{V}$, starting with bos and ending with eos:

$$\mathcal{Y} = \{(\text{bos}, y_1, y_2, \ldots, y_n, \text{eos})|n \geq 0, y_i \in \mathcal{V}\}. \tag{1}$$

The context $x$ can be any modality, such as text (i.e., $x \in \mathcal{Y}$), image, and audio. The tasks include important real-world problems such as machine translation, image captioning, and ASR, where the goal is to produce an output sequence that is appropriate given the input.

A straightforward solution is a maximum a posteriori (MAP) estimate, which selects the most likely output sequence given the input context:

$$\hat{y}_{\text{MAP}} = \arg \max_{y \in \mathcal{Y}} P(y|x). \tag{2}$$

Given that $\mathcal{Y}$ is typically very large in text generation tasks, it is often infeasible to enumerate all possible output sequences in $\mathcal{Y}$. Thus, local optimal search methods such as beam search are used to approximate the MAP estimate as the language models are typically modeled by a autoregressive models (Vaswani et al.,

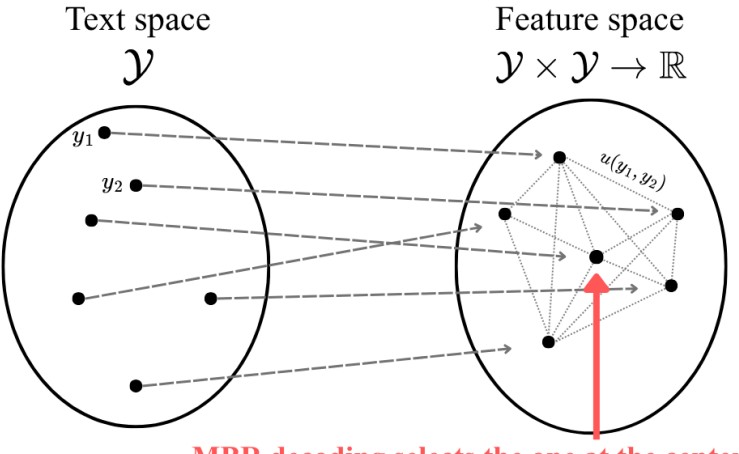

Figure 2: Illustrative explanation of the intuition of the MBR decoding. The hypothesis that lies at the center of the sampled hypotheses is selected as the output. The distance between two hypotheses is inversely related to their utility.

2017). However, MAP decoding, including beam search, is known to generate undesirable outputs, such as an empty sequence, a sequence with repeated tokens, or low-quality text (Wiseman et al., 2017; Holtzman et al., 2020; Eikema & Aziz, 2020). Thus, alternative decoding algorithms have been investigated to improve the quality of the generated text.

## 2.2 Minimum Bayes Risk (MBR) Decoding

MBR decoding works by sampling multiple hypotheses from the model and selecting the one that maximizes the expected utility compared to the rest of the hypotheses (Goel & Byrne, 2000; Kumar & Byrne, 2004; Eikema & Aziz, 2022):

$$\hat{y} = \arg\max_{y \in H} \frac{1}{N} \sum_{y' \in H} u(y, y'), \tag{3}$$

where $H$ is the set of hypotheses sampled from the model, $N = |H|$ is the number of hypotheses, and $u(y, y')$ is a utility function that measures the quality of hypothesis $y$ by treating $y'$ as a pseudo-reference drawn from the set of peer hypotheses. Intuitively, MBR decoding selects the hypothesis that lies at the *center* of the sampled hypotheses, where the *distance* between two hypotheses is inversely related to their utility (Figure 2).[1] It should be noted, however, that this "center" reflects the model's own distribution: if the model generates systematically biased outputs (e.g., a tendency toward shorter sentences), MBR will select a hypothesis near the center of that biased distribution and thus will not correct for such model-intrinsic biases. Whereas MAP decoding selects the sequence with the highest probability in the discrete hypothesis space, MBR decoding selects the one that lies near the middle of the continuous space defined by the utility function. The utility function implicitly defines a continuous space over the hypotheses by quantifying their pairwise similarities.

Previous studies have shown that the way hypotheses $H$ are sampled is crucial for MBR decoding performance (Eikema & Aziz, 2022; Suzgun et al., 2023; Jinnai et al., 2024; Ohashi et al., 2024). Originally, Goel & Byrne (2000) proposed MBR decoding for ASR using beam search to generate $H$, and this was later applied to machine translation (Kumar & Byrne, 2004). However, recent work has found that using unbiased samples drawn from the model is more effective than using beam search for generating $H$ (Eikema & Aziz, 2022). Other studies have also reported that probabilistic sampling methods, such as ancestral sampling, nucleus

---

[1]The utility functions used in MBR decoding are often not symmetric and may not satisfy the triangle inequality, so they are not proper distance functions. Nevertheless, the intuition still holds in many practical cases.

sampling (Holtzman et al., 2020), and epsilon sampling (Hewitt et al., 2022; Freitag et al., 2023), work better than beam search (Eikema & Aziz, 2022; Ohashi et al., 2024).

Another advantage of MBR decoding is that it has theoretical guarantees (Ichihara et al., 2025a). Under mild assumptions, the expected utility of the output chosen by MBR decoding improves as the number of sampled hypotheses increases, with a rate of $O(\frac{1}{\sqrt{N}})$. This result is consistent with empirical findings showing that larger sample sizes lead to higher generation quality (Freitag et al., 2023). In contrast, beam search lacks non-vacuous theoretical guarantees regarding output quality.

The drawback of MBR decoding is its computational cost. The complexity is $O(UN^2 + GN)$, where $U$ is the cost of computing the utility function and $G$ is the cost of generating a hypothesis (Eikema & Aziz, 2022). There are faster algorithms (Cheng & Vlachos, 2023; Deguchi et al., 2024; Trabelsi et al., 2024) that reduce the cost to $O(UN \log N + GN)$ (Jinnai & Ariu, 2024), but this is still much higher than beam search, which is $O(GB)$, where $B$ is the beam width. Note that $G$ represents the computational cost of a full decoding step per hypothesis including any pruning operations as constant factors.

In summary, MBR decoding is a strong alternative to beam search for text generation tasks, and it consistently performs better in many settings. It is not only effective in practice but also has theoretical support. Its main weakness is its computational cost, which makes it less suitable for real-time use. There has been little evaluation of MBR decoding for speech-to-text tasks, which this paper aims to address.

## 3 Related Work

MBR decoding can be understood as a generalized form of *consensus decoding* as it selects the hypothesis that minimizes expected risk with respect to the distribution of sampled hypotheses, effectively finding the consensus of the hypotheses. When the utility function is a token-level edit metric (such as WER or Levenshtein distance), MBR reduces to a form of ROVER-style majority voting (Fiscus, 1997), which directly optimizes for word error rate rather than maximizing the posterior probability (i.e., MAP decoding). Consensus decoding, in turn, can be understood as an instance of reranking methods (Morbini et al., 2012; Chiu & Chen, 2021; Xu et al., 2022; Nakano et al., 2022; Ichihara et al., 2025b) which rank the hypotheses according to its utility and select the best one. Several reranking methods have been proposed for speech recognition using quality estimation (Negri et al., 2014; Ng et al., 2015; Ali & Renals, 2018; Yuksel et al., 2023; Waheed et al., 2025), perplexity (Salazar et al., 2020), deliberation models (Hu et al., 2020; Xu et al., 2022), LLMs (Nie et al., 2022; Hu et al., 2024; Tur et al., 2024), and speech-text foundational models (Shivakumar et al., 2025). The advantage of MBR decoding compared to these approaches is that it does not require any additional training, making it easy to apply to new systems and languages.

Model fusion is another approach to improve the accuracy of ASR and ST systems by combining multiple models (Parikh et al., 2024). This approach has been shown to be effective in various settings, such as combining acoustic models and language models (Lei et al., 2023; Chen et al., 2024) and combining multiple ASR systems (Fiscus, 1997; Tan et al., 2020; Kamo et al., 2025). MBR decoding can be seen as a form of model fusion. In fact, several studies have proposed using MBR decoding to ensemble the outputs from multiple systems (Xu et al., 2010; 2011). At the same time, model fusion can be seen as complementary to MBR decoding, as it focuses on improving the underlying model rather than the decoding process.

Post-editing and error correction are alternative approaches that have been proposed to further improve the accuracy of ASR and speech translation outputs (Liu et al., 2020; Kamiya et al., 2021; Leng et al., 2021; Yang et al., 2023; Ma et al., 2023; Radhakrishnan et al., 2023; Chen et al., 2023). These approaches use language models to correct errors in the initial hypotheses, generating a new hypothesis using the language model (Guo et al., 2019; Hrinchuk et al., 2020; Radhakrishnan et al., 2023). This approach is orthogonal to MBR decoding, as it focuses on refining the output after generation rather than re-evaluating multiple hypotheses during decoding.

| Model | whisper-small | | | whisper-medium | | | whisper-large-v3 | | |
| Metric | WER↓ | MetricX↓ | SemDist↓ | WER↓ | MetricX↓ | SemDist↓ | WER↓ | MetricX↓ | SemDist↓ |
|---|---|---|---|---|---|---|---|---|---|
| Beam ($B = 1$) | 0.067 | 2.091 | 0.085 | 0.087 | 1.818 | 0.073 | 0.036 | 1.744 | 0.064 |
| Beam ($B = 5$) | 0.075 | 2.084 | 0.085 | 0.078 | 1.832 | 0.075 | 0.037 | 1.731 | 0.064 |
| Beam ($B = 20$) | 0.085 | 2.069 | 0.086 | 0.059 | 1.833 | 0.075 | 0.036 | 1.743 | 0.062 |
| MBR ($N = 4$) | 0.054 | 1.923 | 0.074 | 0.044 | 1.693 | 0.066 | 0.031 | 1.660 | 0.058 |
| MBR ($N = 8$) | 0.051 | 1.875 | 0.070 | 0.039 | 1.647 | 0.061 | 0.030 | 1.638 | 0.056 |
| MBR ($N = 16$) | 0.050 | 1.837 | 0.068 | 0.039 | 1.627 | 0.061 | 0.029 | 1.643 | 0.055 |
| MBR ($N = 32$) | 0.050 | 1.823 | 0.068 | **0.039** | 1.626 | 0.060 | 0.029 | 1.631 | 0.053 |
| MBR ($N = 64$) | **0.049** | **1.815** | **0.066** | 0.040 | **1.625** | **0.059** | **0.029** | **1.631** | **0.053** |

| Model | distil-large-v3.5 | | | s2t-small-librispeech-asr | | | seamless-m4t-v2-large | | |
| Metric | WER↓ | MetricX↓ | SemDist↓ | WER↓ | MetricX↓ | SemDist↓ | WER↓ | MetricX↓ | SemDist↓ |
|---|---|---|---|---|---|---|---|---|---|
| Beam ($B = 1$) | 0.040 | 1.838 | 0.056 | 0.045 | 2.258 | 0.042 | 0.035 | 1.850 | 0.031 |
| Beam ($B = 5$) | 0.039 | 1.841 | 0.055 | 0.043 | 2.202 | 0.039 | 0.037 | 1.833 | 0.029 |
| Beam ($B = 20$) | 0.038 | 1.844 | 0.057 | **0.042** | **2.201** | 0.039 | 0.063 | **1.833** | 0.029 |
| MBR ($N = 4$) | 0.035 | 1.774 | 0.049 | 0.051 | 2.411 | 0.046 | 0.040 | 1.926 | 0.033 |
| MBR ($N = 8$) | 0.033 | 1.757 | 0.047 | 0.047 | 2.304 | 0.042 | 0.037 | 1.880 | 0.030 |
| MBR ($N = 16$) | 0.033 | 1.753 | 0.045 | 0.045 | 2.267 | 0.040 | 0.035 | 1.866 | 0.029 |
| MBR ($N = 32$) | 0.033 | 1.749 | 0.045 | 0.043 | 2.287 | 0.040 | 0.035 | 1.864 | 0.029 |
| MBR ($N = 64$) | **0.033** | **1.749** | **0.044** | **0.042** | 2.281 | **0.040** | **0.034** | 1.867 | **0.029** |

Table 1: Evaluation of beam search and MBR decoding on the full LibriSpeech Clean test set with six models. No noise is synthesized in the audio.

## 4  Experiments

The goal of the study is to evaluate MBR decoding for ASR and ST tasks, compared to beam search. We investigate various settings, including different models, datasets, and levels of noise added to the input audio.

**Method.**  We conduct experiments to evaluate the performance of MBR decoding and beam search on various ASR and speech translation tasks. For evaluating the methods under noise, we use the free-sound subset of the Musan dataset (Snyder et al., 2015) to induce background noise to the audio. We sample a noise randomly from the freesound subset of the dataset and crop it to match the length of the input audio. The cropped noise audio is synthesized to the speech with the level of Signal-to-Noise Ratio (SNR) set to 0 dB, noted otherwise. The same noise is used for all the decoding algorithms for fair comparison.

For beam search, we run with a beam width of 1, 5, and 20. We generate up to 64 samples for MBR decoding as hypotheses using Epsilon sampling (Hewitt et al., 2022; Freitag et al., 2023) with $\epsilon = 0.01$ and a temperature set to 1.0. We use the BLEU score (Papineni et al., 2002) implemented by the sacrebleu package (Post, 2018) as the utility function of MBR. We do not use WER (CER) as the utility function because MBR decoding is known to inflate the score used as the utility function which may not accurately reflect a model's true capabilities (Freitag et al., 2022; Kovacs et al., 2024). BLEU scores are computed on the normalized texts using Whisper's normalizer for English (Radford et al., 2023) and `neologdn` normalizer for Japanese (Sato et al., 2017) to avoid unnecessary penalization on punctuation. We use MeCab tokenizer (Kudo, 2005) to tokenize Japanese text for computing the BLEU score.

**Implementation.**  All the code of the experiments is implemented by Python 3 using Huggingface's `transformers` library (Wolf et al., 2020). The experiments are conducted on Linux Ubuntu 22.04 using NVIDIA A100 GPUs. While the codebase is not optimized for efficiency, we report the walltime with our implementation as a reference in Section A.

**Decoding configuration.**  For beam search, we use the standard implementation provided in the HuggingFace's `transformers` library. The beam width specifies the number of active hypotheses maintained at each decoding step. No additional pruning strategies are applied: in particular, we do not use threshold

| Dataset | LibriSpeech | | | VoxPopuli | | | AMI-IHM | | |
|---|---|---|---|---|---|---|---|---|---|
| Metric | WER↓ | MetricX↓ | SemDist↓ | WER↓ | MetricX↓ | SemDist↓ | WER↓ | MetricX↓ | SemDist↓ |
| Beam ($B = 1$) | 0.081 | 2.250 | 0.091 | 0.117 | 1.500 | 0.067 | **0.380** | 2.280 | **0.264** |
| Beam ($B = 5$) | 0.081 | 2.230 | 0.092 | 0.117 | 1.500 | 0.067 | **0.380** | 2.280 | **0.264** |
| Beam ($B = 20$) | 0.082 | 2.200 | 0.090 | 0.117 | 1.500 | 0.067 | **0.380** | 2.280 | **0.264** |
| MBR ($N = 64$) | **0.057** | **2.000** | **0.077** | **0.098** | **1.400** | **0.053** | 0.568 | **2.200** | 0.284 |

Table 2: Evaluation of beam search and MBR decoding on English ASR tasks with Whisper-large-v3. Noise is synthesized to the audio. The signal-to-noise ratio is 0 dB.

| Dataset | ReazonSpeech | | | CommonVoice | | | JSUT | | |
|---|---|---|---|---|---|---|---|---|---|
| Metric | CER↓ | MetricX↓ | SemDist↓ | CER↓ | MetricX↓ | SemDist↓ | CER↓ | MetricX↓ | SemDist↓ |
| Beam ($B = 1$) | 0.305 | 2.975 | 0.143 | 0.306 | 2.825 | 0.134 | 0.183 | 2.250 | 0.088 |
| Beam ($B = 5$) | 0.307 | 2.975 | 0.143 | 0.302 | 2.875 | 0.132 | 0.185 | 2.350 | 0.089 |
| Beam ($B = 20$) | 0.308 | 3.050 | 0.140 | 0.306 | 2.875 | 0.133 | 0.184 | 2.350 | 0.090 |
| MBR ($N = 64$) | **0.291** | **2.875** | **0.130** | **0.297** | **2.725** | **0.123** | **0.177** | **2.200** | **0.082** |

Table 3: Evaluation of decoding methods on Japanese ASR tasks with Kotoba-Whisper. Noise is synthesized to the audio. The signal-to-noise ratio is 0 dB.

pruning (discarding hypotheses based on a score gap from the best path) or diversity penalties.[2] All reported beam-search results therefore reflect the library defaults beyond the beam width itself.

## 4.1 Automatic Speech Recognition (ASR)

**Resources.** We evaluate the performance of MBR decoding on ASR using LibriSpeech (clean) (Panayotov et al., 2015), AMI-IHM (Carletta, 2007), and VoxPopuli (Wang et al., 2021a) for English, ReazonSpeech (Yin et al., 2023), Common Voice-v8 (Ardila et al., 2020), and JSUT (Sonobe et al., 2017) for Japanese. We use Whisper (Radford et al., 2023)[3] for English and Kotoba-Whisper-v2[4] for Japanese ASR models. All the audio files are resampled to 16 kHz to meet the Whisper model's requirement. For the results on Table 1, we use the full test set of LibriSpeech dataset, which contains 2,620 samples, including those longer than 30 seconds. For samples exceeding 30 seconds, because Whisper models do not natively handle audio longer than 30 seceonds, we apply the sequential long-form algorithm provided by the Whisper model (Chiu et al., 2019; Narayanan et al., 2019; Koluguri et al., 2024). For the other experiments, we use the first 1000 samples in the test set for the evaluation, excluding samples longer than 30 seconds to isolate the effect of MBR decoding from any interaction with long-form handling techniques. Most of the samples are shorter than 30 seconds, and the exclusion rate is negligible across these datasets (e.g., only 9 out of 2620 are longer than 30 seconds in LibriSpeech).

**Evaluation metrics.** We use word error rate (WER) for English and character error rate (CER) for Japanese as the main evaluation metrics. The same normalizers as BLEU scores are used for WER (whisper normalizer) and CER (`neologdn` normalizer). In addition, SemDist (Kim et al., 2021) and MetricX (`metricx-23-xxl-v2p0`; Juraska et al. 2023) are used to evaluate the semantic similarity and overall quality of the generated outputs. SemDist is a metric that measures the semantic distance between the generated text and the reference text using the inner product of the embeddings of the texts, which is also known as other names such as cosine distance and contextual similarity in the NLP community (Akula & Garibay, 2022; Mukherjee & Shrivastava, 2022). It is proposed to complement the problem of WER (CER), which does not capture semantic similarity well, and thus, the effectiveness of the generation in the downstream tasks is not clear by itself. We use a sentence BERT model named `all-MiniLM-L6-v2` as the embedding

---

[2]Specifically, it follows the implementation and default parameters of `transformers` version 4.55.0. `https://github.com/huggingface/transformers/blob/v4.55.0/src/transformers/models/whisper/generation_whisper.py`

[3]`https://huggingface.co/openai/whisper-large-v3`

[4]`https://huggingface.co/kotoba-tech/kotoba-whisper-v2.0`

| SNR (dB) | -20 | -15 | -10 | -5 | 0 | 5 | 10 | 15 | 20 |
|---|---|---|---|---|---|---|---|---|---|
| Beam ($B = 1$) | 0.590 | 0.458 | 0.290 | 0.143 | 0.081 | 0.055 | 0.049 | 0.049 | 0.045 |
| Beam ($B = 5$) | 0.599 | 0.446 | 0.293 | 0.152 | 0.081 | 0.056 | 0.048 | 0.049 | 0.045 |
| Beam ($B = 20$) | 0.590 | 0.444 | 0.284 | 0.151 | 0.082 | 0.056 | 0.049 | 0.048 | 0.045 |
| MBR ($N = 64$) | **0.530** | **0.388** | **0.235** | **0.108** | **0.057** | **0.041** | **0.035** | **0.036** | **0.034** |

Table 4: WER scores on the LibriSpeech dataset with different SNR levels of the speech compared to the synthesized noise.

| SNR (dB) | 0 | 5 | 10 | 15 | 20 |
|---|---|---|---|---|---|
| Beam ($B = 1$) | 0.305 | 0.273 | 0.258 | 0.243 | 0.243 |
| Beam ($B = 5$) | 0.307 | 0.282 | 0.254 | 0.243 | 0.240 |
| Beam ($B = 20$) | 0.308 | 0.272 | 0.254 | 0.242 | 0.235 |
| MBR ($N = 64$) | **0.291** | **0.250** | **0.238** | **0.229** | **0.223** |

Table 5: CER scores on the ReazonSpeech dataset with different SNR levels of the speech compared to the synthesized noise.

model to compute SemDist (Reimers & Gurevych, 2019).[5] MetricX is one of the state-of-the-art metrics for machine translation that evaluates the overall quality of the generated outputs by learning human MQM evaluation results. We use it for assessing the overall quality of the generated outputs.

**Models.** We use whisper-small, whisper-medium, whisper-large-v3, and distil-whisper to evaluate the effect of the model size. Table 1 shows that MBR decoding outperforms beam search in all the model sizes. The result shows that MBR decoding is effective regardless of the model size. We note that beam search results are identical across all tested beam widths ($B = 1, 5, 20$) for the Whisper models evaluated on LibriSpeech (Table 1), VoxPopuli and AMI-IHM (Table 2), and all seven languages in Table 8. We have verified these results against our experimental logs and confirmed their correctness. This behavior stems from the highly peaked probability distributions of the Whisper model family (Radford et al., 2023): the model is extremely confident in its predictions, so the greedy path ($B = 1$) already coincides with the beam-optimal path even for larger beam widths.

We additionally evaluate two non-Whisper sequence-to-sequence models on the LibriSpeech Clean test set: `facebook/s2t-small-librispeech-asr` (S2T; Wang et al. 2020) and `facebook/seamless-m4t-v2-large` (SeamlessM4T; Communication et al. 2023). MBR decoding achieves competitive or better performance than beam search for both models, confirming that the advantage of MBR generalizes across different autoregressive architectures. SeamlessM4T shows larger variation across beam widths (e.g., WER degrades with wider beam), while MBR consistently selects a reliable hypothesis. We also attempted to apply MBR decoding to Wav2Vec 2.0 (Baevski et al., 2020), a CTC-based model. However, the per-frame probability distributions of CTC models are extremely peaked, so random sampling yields the same result as greedy (MAP) decoding unless the temperature is raised to a level that severely degrades output quality. We therefore conclude that MBR decoding is most effective for models with sufficiently diverse output distributions.

**Number of samples for MBR decoding.** Table 1 shows the performance of MBR decoding with different numbers of samples. Surprisingly, with only four to eight samples, MBR decoding outperforms beam search. The result shows that MBR decoding is effective even with a small amount of additional computation, which might be admissible for real-time ASR tasks. Still, we observe that the accuracy of MBR decoding improves with a larger number of samples, suggesting that more computation can lead to better performance.

---

[5]`https://huggingface.co/sentence-transformers/all-MiniLM-L6-v2`

| Metric | WER↓ | MetricX↓ | SemDist↓ |
|---|---|---|---|
| Beam $(B = 1)$ | 0.042 | 1.750 | 0.060 |
| MBR $(N = 64, u = \text{BLEU})$ | **0.033** | **1.650** | 0.053 |
| MBR $(N = 64, u = \text{BLEURT})$ | 0.035 | **1.650** | 0.056 |
| MBR $(N = 64, u = \text{SentBERT})$ | 0.034 | 1.675 | **0.050**[*] |

Table 6: Evaluation of MBR decoding with varying utility functions on the LibriSpeech dataset with whisper-large-v3. No noise is synthesized in the audio. [*]MBR using SentBERT may lead to inflate SemDist scores that do not accurately reflect a model's true capabilities (Kovacs et al., 2024).

| Metric | WER↓ | MetricX↓ | SemDist↓ |
|---|---|---|---|
| Beam $(B = 1)$ | 0.042 | 1.750 | 0.060 |
| MBR $(N = 64, \epsilon = 0)$ | 0.033 | **1.625** | 0.052 |
| MBR $(N = 64, \epsilon = 0.01)$ | 0.033 | 1.650 | 0.053 |
| MBR $(N = 64, \epsilon = 0.02)$ | **0.033** | 1.650 | **0.052** |

Table 7: Evaluation of MBR decoding with varying sampling parameters on the LibriSpeech dataset with whisper-large-v3. No noise is synthesized in the audio.

**Correlation of MBR objective values to error rates.** To investigate how much the MBR objective indicates the utility of the given hypothesis, we compute the correlation of the MBR objective with the WER. Pearson correlation coefficient is computed for each instance of the LibriSpeech with no synthesized noise over 64 samples generated by whisper-large-v3. Then, we estimate it with the average over the 1000 instances. The average value of the Pearson correlation coefficient is -0.3913, and the standard error is 0.0129, indicating that the MBR objective has a substantial negative correlation with the target objective (negative correlation because MBR objective is higher the better, and WER is lower the better). This suggests that it is a reasonable approach to use it as the reranking procedure for ASR.

**Datasets.** Tables 2 and 3 show the performance of the decoding algorithms using WER (CER), SemDist, and MetricX. MBR decoding outperforms beam search in all the datasets except for AMI-IHM, suggesting that the advantage of MBR decoding over beam search is in a wide range of domains and on both lexical and semantic levels.

**Speech length.** Given that MBR decoding fails to improve on the AMI-IHM dataset, we conduct a post-hoc error analysis. The corpus records all meeting utterances, resulting in many very short transcriptions that contain non-lexical fillers (e.g., *yeah*, *hmm*, *gosh*). To investigate MBR's performance on these instances, we compute the average WER of beam search and MBR decoding, split by the number of words in the reference transcription (Figure 3). While MBR decoding has a comparable WER to beam search overall, it shows a significantly higher WER on instances shorter than six words. One of the reasons is likely due to the limitations of BLEU on short sequences. If a single filler token is missed or substituted (e.g., *yeah* vs. *yes*), BLEU yields zero overlap, creating a flat utility landscape where no hypothesis is clearly distinguished. In this case MBR objective is not informative for selecting the hypothesis, and thus, MBR decoding fails to improve the performance over beam search.

**Noise level.** Tables 4 and 5 show the performance under different noise levels. The result shows that MBR decoding is more accurate than beam search at any noise level.

**Utility functions for MBR decoding.** The performance of MBR decoding is known to be dependent on the choice of the utility function (Freitag et al., 2022; Kovacs et al., 2024). We evaluate MBR decoding using SentBERT (Reimers & Gurevych, 2019; 2020) and BLEURT (`BLEURT-20-D12`) in addition to using BLEU. SentBERT is a sentence-level embedding model that captures semantic similarity between sentences

| Domain | Arabic | Chinese | Hindi | Indonesian | Tamil | Thai | Vietnamese |
|---|---|---|---|---|---|---|---|
| Beam ($B = 1$) | 0.305 | 0.231 | 0.180 | 0.090 | 0.278 | 0.366 | 0.193 |
| Beam ($B = 5$) | 0.305 | 0.231 | 0.180 | 0.090 | 0.278 | 0.366 | 0.193 |
| Beam ($B = 20$) | 0.305 | 0.231 | 0.180 | 0.090 | 0.278 | 0.366 | 0.193 |
| MBR ($N = 64$) | **0.259** | **0.205** | **0.142** | **0.069** | **0.260** | **0.354** | **0.180** |

Table 8: WER (CER) scores of MBR decoding and beam search on the CommonVoice dataset with Whisper-large-v3. No noise is synthesized in the audio. CER is reported for Chinese and WER for other languages.

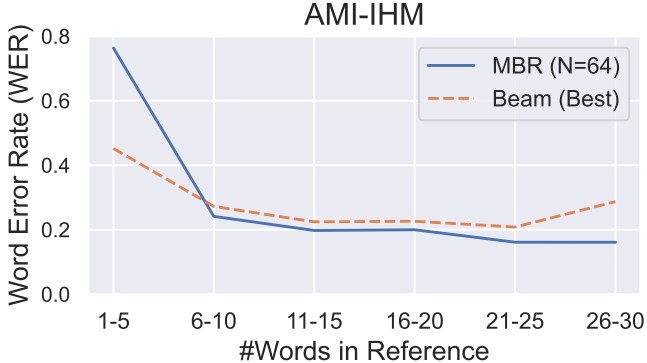

Figure 3: WER of AMI-IHM averaged over the instances with the number of words in the reference text is in the range of (x, x+5].

computed by the cosine similarity between the two embedding vectors. Thus, the value is 1 minus the value of SemDist. We use the `all-MiniLM-L6-v2` model as the embedding model to compute SentBERT. Table 6 shows that the differences in accuracy using these utility functions are marginal, yet they all outperform beam search. The result shows that the advantage of MBR decoding over beam search is robust to the choice of the utility function. SentBERT achieves the best SemDist score, which is expected as it is directly optimized for the metric (Freitag et al., 2022).

**Sampling algorithm for MBR decoding.** The choice of sampling algorithm is known to be crucial for the performance of MBR decoding in machine translation tasks (Freitag et al., 2023; Ohashi et al., 2024; Jinnai et al., 2024). We evaluate epsilon sampling with varying epsilon values of 0.00, 0.01, and 0.02. Table 7 shows the performance of MBR decoding with the different epsilon values. The result shows that the performance of MBR decoding is relatively robust to the choice of epsilon values, and it outperforms beam search in all the settings. It also indicates that the effective sampling strategy for ASR may be different from the effective strategy for machine translation (i.e., epsilon sampling), which may be an interesting avenue of future work. We attribute this robustness to the high-confidence nature of current ASR models: in ASR, the model is constrained to transcribe specific acoustic features, resulting in a sharply peaked output distribution where a small number of tokens dominate the probability mass. This makes sampling parameters such as $\epsilon$ less consequential than in machine translation, where greater lexical diversity (synonyms, rephrasing) leads to a flatter distribution. This observation is likely tied to the choice of model and task domain, and may not hold universally across all ASR systems.

**Languages.** To assess whether the performance of MBR decoding is language-specific or generic to natural language text generation tasks, we conduct experiments on the following languages: Arabic (ar), simplified Chinese (zh-CN), Hindi (hi), Tamil (ta), Thai (th), and Vietnamese (vi). We use the test split of the CommonVoice-v8 dataset and evaluate the WER (CER for Chinese). We use spaCy-Thai for segmenting

|  | LibriSpeech | ReazonSpeech |
|---|---|---|
| Beam ($B = 1$) | 0.042 | 0.305 |
| NoRefER ($N = 64$) | 0.073 | 0.368 |
| ProGRes ($N = 64$) | 0.043 | 0.358 |
| MBR ($N = 64$) | **0.033** | **0.291** |
| Oracle ($N = 64$) | 0.013 | 0.149 |

Table 9: WER (CER) of the reranking algorithms.

| Domain | CoVoST2 (Ja-En) | | | | FLEURS (Ja-En) | | | |
|---|---|---|---|---|---|---|---|---|
| Metric | BLEU↑* | ROUGE-L↑ | BLEURT↑ | MetricX↓ | BLEU↑* | Rouge-L↑ | BLEURT↑ | MetricX↓ |
| Beam ($B = 1$) | 18.646 | 42.283 | -0.184 | 2.825 | 6.218 | 30.178 | -0.486 | 6.750 |
| Beam ($B = 5$) | 18.685 | 41.825 | -0.201 | 2.850 | 6.158 | 29.954 | -0.487 | 6.725 |
| Beam ($B = 20$) | 18.122 | 41.362 | -0.235 | 2.950 | 6.202 | 29.886 | -0.489 | 6.825 |
| MBR ($N = 64$) | **22.456** | **47.572** | **-0.073** | **2.475** | **8.078** | **34.212** | **-0.365** | **6.100** |
| Domain | CoVoST2 (En-Ja) | | | | FLEURS (En-Ja) | | | |
| Metric | BLEU↑* | Rouge-L↑ | BLEURT↑ | MetricX↓ | BLEU↑* | Rouge-L↑ | BLEURT↑ | MetricX↓ |
| Beam ($B = 1$) | 10.395 | 34.176 | 0.110 | 4.975 | 8.242 | 30.771 | -0.015 | 8.975 |
| Beam ($B = 5$) | 10.795 | 33.960 | 0.109 | 4.950 | 8.360 | 30.612 | -0.019 | 8.950 |
| Beam ($B = 20$) | 10.822 | 34.393 | 0.116 | 4.900 | 8.224 | 30.537 | -0.015 | 8.900 |
| MBR ($N = 64$) | **15.968** | **43.260** | **0.195** | **4.225** | **11.681** | **37.207** | **0.017** | **6.375** |

Table 10: Evaluation of decoding algorithms on speech translation. *BLEU scores are used as the utility function for MBR decoding, which may lead to artificially inflated scores that do not accurately reflect a model's true capabilities (Kovacs et al., 2024).

words in Thai (Zeman et al., 2017).[6] Table 8 shows the result. Overall, we observe MBR decoding to consistently outperform beam search in all the languages. The result indicates that the method is effective across different languages.

**Comparison to reranking decoding algorithms.** In contrast to MBR decoding which selects the hypothesis that has the highest agreement with the other hypotheses, reranking algorithms rescore a fixed set of hypotheses using an external scoring model. We evaluate two reranking algorithms proposed recently. NoRefER selects the sentence with highest score according to a language model fine-tuned for the ASR reranking task (Yuksel et al., 2023).[7] NoRefER does not use the audio input on reranking and relies solely on the generations.

ProGRes selects the hypothesis using the weighted sum of the two objectives, LLM score and ASR score (Tur et al., 2024). LLM score is the perplexity of the hypothesis given a prompt articulated for the reranking task as a context $c$. We use the same prompt as in Section 2.1 of Tur et al. (2024). Tur et al. (2024) evaluate ProGRes using Llama-3, GPT-3.5, GPT-4 and show that GPT-4 achieves the best performance over the three. Unfortunately, the logits of GPT-3.5 and GPT-4 are no longer provided to the users, so it is not reproducible using these proprietary models. To this end, we use Llama-3 for computing the LLM score in the following experiment (Grattafiori et al., 2024).[8] ASR score is the loss value of the ASR model. We use cross-entropy loss, one of the standard loss functions for ASR models, as the loss function for Whisper is not disclosed (Radford et al., 2023). We set the weight of the LLM score to $\alpha = 0.05$ as it performs the best in the experiments by Tur et al. (2024).

---

[6]https://pypi.org/project/spacy-thai/
[7]https://huggingface.co/aixplain/NoRefER
[8]https://huggingface.co/meta-llama/Meta-Llama-3-8B-Instruct

Table 9 shows the comparison of the reranking algorithms on LibriSpeech and ReazonSpeech. Overall, we observe the performance of the algorithms to be suboptimal compared to MBR decoding and beam search. NoRefER is trained to distinguish models compressed into different sizes so that they have sufficiently different accuracy (Yuksel et al., 2023). Thus, it may be less effective for reranking samples from the same model.

The Oracle score is the score of the hypothesis with the lowest WER (CER) to the reference in the 64 hypotheses sampled. Given that it achieves significantly better score than any of the reranking algorithms, the hypotheses set has good enough hypothesis to be selected and the reranking algorithms have room of improvement.

## 4.2 Speech Translation

We use the English and Japanese subsets of CoVoST2 (Wang et al., 2021b) and FLEURS (Conneau et al., 2023) datasets for speech translation. We use Kotoba-Whisper-Bilingual for speech translation system.[9] Kotoba-Whisper-Bilingual is a model fine-tuned on top of the distilled Whisper model and trained on a large amount of bilingual speech translation data. It is one of the state-of-the-art open-source systems for bilingual speech recognition and translation for English and Japanese.

We use BLEU using sacrebleu, ROUGE-L (Lin, 2004), BLEURT (Sellam et al., 2020), and MetricX as the evaluation metrics. The other settings are the same as the ASR. Table 10 shows the results of the experiments. Overall, MBR decoding outperforms beam search in all the metrics in both language pairs and datasets.

Note that MBR decoding tends to achieve a relatively higher score than the others on the utility function used during the decoding process (Freitag et al., 2022), which may be indicative of overfitting. Thus, BLEU scores in Table 10 should be interpreted as references.

## 5 Conclusions

In this paper, we empirically evaluate the performance of MBR decoding for offline ASR and ST tasks. We compare MBR decoding and beam search on a wide range of scenarios with various models, languages, datasets, noise levels, evaluation metrics, and hyperparameters. The extensive evaluation shows that MBR decoding consistently achieves higher accuracy than beam search in both speech-to-text tasks.

The results indicate that MBR decoding has the potential to improve the state-of-the-art performance of offline speech-to-text tasks. Unlike other approaches that depend on heuristics, MBR decoding has a theoretical guarantee (Ichihara et al., 2025a). We believe that MBR decoding is a promising approach for a wide range of speech-to-text tasks and should be considered as one of the baseline methods to improve the system accuracy.

### Broader Impact Statement

MBR decoding incurs a computational complexity of $O(UN^2 + GN)$, substantially higher than the $O(GB)$ cost of beam search. When deployed at scale for large-scale transcription tasks, this increased computational burden translates directly into higher energy consumption and carbon emissions. We therefore advise practitioners to measure the computational footprint before deploying MBR in production. A practical strategy to reduce unnecessary computation is the *doubling trick* (Besson & Kaufmann, 2018; Jinnai & Ariu, 2024). Doubling trick starts with a small number of samples, iteratively double the count, and terminate once the selected hypothesis stabilizes between iterations (e.g., the same hypothesis is selected twice in a row). This approach allows practitioners to dynamically bound the computation while retaining the accuracy benefits of MBR decoding.

---

[9]https://huggingface.co/kotoba-tech/kotoba-whisper-bilingual-v1.0

**Acknowledgments**

We would like to thank the Action Editor and the reviewers for their constructive feedback to the manuscript. We are also grateful for the constructive feedback and insightful conversation by the colleagues and fellow researchers which helped shape the research question.

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

## A    Limitations

One of the critical limitations of MBR decoding is the computational cost. For the sake of reference, we provide the walltime of the decoding algorithms with our implementation (Appendix B).

There are several libraries dedicated to optimizing the speed of the whisper models, such as faster-whisper[10] and whisper.cpp.[11] Thus, beam search can be made faster by using such libraries, but MBR decoding may require additional modifications to fully leverage these optimizations. Developing a fast implementation of MBR decoding is left for future work.

Experiments are primarily conducted using sequence-to-sequence autoregressive models (Vaswani et al., 2017; Radford et al., 2023). We also attempted to apply MBR decoding to Wav2Vec 2.0 (Baevski et al., 2020), a CTC-based model. CTC models produce extremely peaked per-frame probability distributions: random sampling yields the same result as greedy (MAP) decoding unless the temperature is raised to a level that severely degrades output quality. Consequently, MBR decoding is not directly applicable to CTC-based models in their standard configuration. Evaluation of MBR decoding to wider range of models remains an open direction for future work.

The Musan dataset (Snyder et al., 2015) covers a wide range of noise types, but it may not fully represent the noise encountered in all the communities and regions. Evaluation using real-world noisy datasets for the particular communities and regions is left for future work.

## B    Walltime

Table 11 shows the average walltime on the LibriSpeech Clean dataset with the whisper-large-v3 model. Note that because the experiment is not conducted to evaluate the walltime of the decoding algorithms, our codebase is not optimized to reduce the walltime. For example, the reported values include the time for logging and sending the generated hypotheses to a cloud server, which adds to the overall time. Also note that the walltime also depends on the choice of the utility function. Currently, computing the BLEU scores on CPU is taking the majority of the computation time. We find that using SentBERT as the utility function is much faster than using BLEU, as SentBERT runs on a GPU in parallel and does not require CPU/GPU data transfer. Thus, the reported time does not reflect the performance of optimized implementations and should solely be considered as a reference.

---

[10]`https://github.com/systran/faster-whisper`
[11]`https://github.com/ggml-org/whisper.cpp`

| Method | Walltime (seconds) | WER |
|--------|--------------------|-----|
| Beam $(B = 1)$ | 0.88 | 0.042 |
| Beam $(B = 5)$ | 1.54 | 0.042 |
| Beam $(B = 20)$ | 1.56 | 0.042 |
| MBR $(N = 4)$ | 2.47 | 0.035 |
| MBR $(N = 8)$ | 3.44 | 0.035 |
| MBR $(N = 16)$ | 7.97 | 0.034 |
| MBR $(N = 32)$ | 17.89 | 0.032 |
| MBR $(N = 64)$ | 30.18 | 0.033 |

Table 11: Estimated average walltime of the decoding algorithms on the LibriSpeech dataset with the Whisper-large-v3 model. Note that the walltime includes the time for logging and sending the generated hypotheses to a cloud server for record, which adds to the overall time. Thus, the reported time does not reflect the performance of optimized implementations and should solely be considered as a reference.

## C   Reproducibility Statement

The code for our experiment is available at `https://github.com/CyberAgentAILab/mbr-for-asr`. All the experiments are conducted using publicly available resources shown in Table 12.

| Datasets | |
|---|---|
| LibriSpeech | `https://huggingface.co/datasets/openslr/librispeech_asr` (Panayotov et al., 2015) |
| VoxPopuli | `https://huggingface.co/datasets/facebook/voxpopuli` (Wang et al., 2021a) |
| AMI-IHM | `https://huggingface.co/datasets/edinburghcstr/ami` (Carletta, 2007) |
| ReazonSpeech | `https://huggingface.co/datasets/japanese-asr/ja_asr.reazonspeech_test` (Yin et al., 2023) |
| CommonVoice-v8 | `https://huggingface.co/datasets/mozilla-foundation/common_voice_8_0` (Ardila et al., 2020) |
| JSUT | `https://huggingface.co/datasets/japanese-asr/ja_asr.jsut_basic5000` (Sonobe et al., 2017) |
| CoVoST2 | `https://huggingface.co/datasets/facebook/covost2` (Wang et al., 2021b) |
| FLEURS | `https://huggingface.co/datasets/google/fleurs` (Conneau et al., 2023) |
| Models | |
| whisper-large-v3 | `https://huggingface.co/openai/whisper-large-v3` (Radford et al., 2023) |
| whisper-small | `https://huggingface.co/openai/whisper-small` (Radford et al., 2023) |
| whisper-medium | `https://huggingface.co/openai/whisper-medium` (Radford et al., 2023) |
| distil-whisper | `https://huggingface.co/distil-whisper/distil-large-v3.5` (Gandhi et al., 2023) |
| kotoba-whisper | `https://huggingface.co/kotoba-tech/kotoba-whisper-v2.0` |
| kotoba-whisper-bilingual | `https://huggingface.co/kotoba-tech/kotoba-whisper-bilingual-v1.0` |
| Others | |
| BLEURT | `https://huggingface.co/lucadiliello/BLEURT-20-D12` (Sellam et al., 2020) |
| MetricX | `https://huggingface.co/google/metricx-23-xxl-v2p0` (Juraska et al., 2023) |
| all-MiniLM-L6-v2 | `https://huggingface.co/sentence-transformers/all-mpnet-base-v2` (Reimers & Gurevych, 2019; 2020) |
| NoRefER | Because only part of the code is published (`https://huggingface.co/aixplain/NoRefER`), the method is implemented by us. (Yuksel et al., 2023) |
| ProGRes | Because only part of the code is published (`https://github.com/AdaDTur/ProGRes`), the method is implemented by us. (Tur et al., 2024) |
| Llama-3 | `https://huggingface.co/meta-llama/Meta-Llama-3-8B-Instruct` (Grattafiori et al., 2024) |

Table 12: List of datasets and models used in this study. All the resources are publicly available.

