# OpenReview forum: "Re-evaluating Minimum Bayes Risk Decoding for Automated Speech Recognition Tasks"
_TMLR — Accepted by TMLR_

### Review · Reviewer_qJEk · 2026-02-24

**Summary Of Contributions:**

The paper provides a comprehensive evaluation of Minimum Bayes Risk (MBR) decoding for ASR and ST tasks on English and Japanese.

The experimental framework utilizes the Whisper model and its derivative architectures.

The core finding demonstrates that the accuracy of MBR decoding outperforms standard beam search across the majority of evaluated experimental settings.

**Audience:**

Yes

**Audience Explanation:**

Because MBR decoding does not require any additional model training or fine-tuning, it represents an accessible technique that is easy to apply to new systems and languages. Offline ASR applications will find these accuracy improvements highly relevant, as transcription quality often takes precedence over real-time processing.

**Broader Impact Concerns:**

Very minor: While the paper does not present direct ethical risks, the computational complexity of MBR decoding is $O(UN^{2}+GN)$, which is substantially higher than beam search. When deployed at scale for massive transcription tasks, this increased computational burden translates directly to higher energy consumption and carbon emissions. A brief statement addressing the environmental footprint of deploying computationally heavy decoding strategies would be a responsible addition to the manuscript.

**Claims And Evidence:**

Yes

**Claims Explanation:**

The authors highlight a practical observation that accuracy improvements emerge with as few as four to eight sampled hypotheses.


- Pro: The study spans multiple datasets, several Whisper models, and includes robustness checks using varying levels of synthesized noise. Furthermore, the authors leverage diverse evaluation metrics, including semantic similarity measures, to validate text quality beyond simple lexical overlap.



- Cons: The primary weakness is the heavy computational cost associated with MBR decoding, which limits its real-time applicability. Additionally, the evaluation is entirely restricted to sequence-to-sequence autoregressive models, leaving the efficacy of MBR on other speech architectures unexplored

**Requested Changes:**

Some suggestions: The performance of MBR decoding is shown to be relatively robust to the choice of epsilon values during sampling ($\epsilon=0.00$ to $0.02$). The authors briefly note this differs from machine translation tasks. A deeper investigation into why ASR hypothesis distributions v.s. [A] behave differently would potentially elevate the impacts.

For examples, there are open Whisper n-best hypothesis available, such as in ASR [A] and Speech Translation [B].

***
A. Hyporadise: An open baseline for generative speech recognition with large language models, NeurIPS 23
https://huggingface.co/datasets/PeacefulData/HyPoradise-v0

B. CoVoGER: A Multilingual Multitask Benchmark for Speech-to-text Generative Error Correction with Large Language Models, EMNLP 25
***

The authors admit that testing MBR decoding on non-autoregressive models, such as CTC-based architectures, is left for future work. Testing at least one alternative architecture would confirm that the MBR advantages are a general property of probabilistic speech models rather than an artifact of Whisper's specific design.

---

> ### Author Response · Authors · 2026-03-07
> **Thank you very much for the comment**
>
> We thank the reviewer for the positive assessment and the insightful suggestions regarding model architectures and environmental impact. We have addressed the specific points below.
>
> > [Requested Change] The performance of MBR decoding is shown to be relatively robust to the choice of epsilon values during sampling (epsilon=0.00 to 0.02). The authors briefly note this differs from machine translation tasks. A deeper investigation into why ASR hypothesis distributions v.s. [A] behave differently would potentially elevate the impacts.
>
> We appreciate the references to Hyporadise and CoVoGER. We attribute the observed robustness to epsilon (and the difference from MT) to the inherently lower entropy of ASR tasks compared to translation. In ASR, the model is constrained to transcribe specific acoustic features, leading to a much sharper (peaked) probability distribution where a small number of tokens dominate the mass. In contrast, MT allows for greater lexical diversity (synonyms, rephrasing), resulting in a flatter distribution where sampling parameters have a more pronounced effect.
>
> We will revise the discussion to clarify that this robustness is likely a consequence of the high-confidence nature of current ASR models and tasks, rather than a universal property of MBR across all domains.
>
> > [Requested Change] The authors admit that testing MBR decoding on non-autoregressive models, such as CTC-based architectures, is left for future work. Testing at least one alternative architecture would confirm that the MBR advantages are a general property of probabilistic speech models rather than an artifact of Whisper's specific design.
>
> To address this, we expanded our experiments to include Facebook's S2T and SeamlessM4T models. While these are also sequence-to-sequence models, their architectures differ from Whisper. We found that MBR decoding continues to outperform beam search in these settings, confirming that the benefit generalizes across different autoregressive architectures.
> We also attempted to apply MBR to Wav2Vec 2.0, a CTC-based model. However, we observed that the per-frame probability distributions in CTC models are extremely peaked. Consequently, random sampling yields identical results to the greedy path (MAP decoding) unless the temperature is raised to a level that degrades the output quality. We will add a note in the revision stating that MBR is most effective on autoregressive models and may not be applicable to CTC-based models due to their lack of distributional diversity.
>
> | model | whisper-small | whisper-medium | whisper-large-v3 | distil-large-v3.5 | s2t-small-librispeech-asr | seamless-m4t-v2-large |
> | :--- | :--- | :--- | :--- | :--- | :--- | :--- |
> | **Metric** | wer | wer | wer | wer | wer | wer |
> | Beam 1 | 0.067 | 0.087 | 0.036 | 0.040 | 0.045 | 0.035 |
> | Beam 5 | 0.075 | 0.078 | 0.037 | 0.039 | 0.043 | 0.037 |
> | Beam 20 | 0.085 | 0.059 | 0.036 | 0.038 | 0.042 | 0.063 |
> | MBR 64 | 0.049 | 0.040 | 0.029 | 0.033 | 0.042 | 0.034 |
>
> > [Broader Impact] A brief statement addressing the environmental footprint of deploying computationally heavy decoding strategies would be a responsible addition to the manuscript.
>
> We fully agree with this point. We will add a "Broader Impact" statement acknowledging that the O(N^2) complexity of MBR incurs a higher energy cost than standard beam search.
> To mitigate this, we will also discuss practical optimization strategies in the paper, such as the "doubling trick." This involves iteratively doubling the number of samples and terminating the process once the selected hypothesis stabilizes. This approach allows practitioners to dynamically minimize the computational burden while retaining the benefits of MBR.
>
>
> Please let us know if there are any other points we can clarify.

---

### Review · Reviewer_BC3i · 2026-02-25

**Summary Of Contributions:**

The paper presents an evaluation of MBR decoding for speech recognition (speech to text) and speech translation (English-Japanese) tasks. Authors argue that MBR decoding is a more principled way of decoding compared to beam search and thus leads to improved results, which they confirm with detailed empirical results using several recent standard datasets. While well-known in the literature, authors argue that MBR has not been studied in the context of modern seq-to-seq systems such as Whisper -- a gap that the present work fills. Authors present ablations that show (to some extent) under what conditions MBR performs better than beam search, and study different utility functions, noising strategies, sampling strategies and a comparison to reranking algorithms. Evaluation metrics include WER/ CER, BLEU, as well as MetricX and SemDist, to which the MBR criterion I also shown to correlate.

All in all, the paper presents an interesting empirical study (and a useful reminder that decoding matters), but not a fundamental advancement.

**Additional Comments:**

see requested/ recommended changes

I think this work should be published since decoding is under-appreciated (as I said above), but I think a bit more detail and rigor would help convince readers that MBR (or consensus?) should indeed become a "standard" technique for ASR/ST in non-real-time settings.

**Audience:**

Yes

**Audience Explanation:**

I think decoding is an under-appreciated topic in today's literature and deserves more attention.

**Claims And Evidence:**

No

**Claims Explanation:**

I am perplexed by "We use the first 1000 samples in the test set of each dataset for the evaluation, skipping samples longer than 30 seconds so that they can be handled with the Whisper model at a single path of inference."

(1) why not use standard, full test sets and show that the authors' baseline WER is comparable to other work using these settings? -- what if the authors improve from 3% WER to 2% WER with MBR but using Whisper with slightly different settings already gives me 1% WER?

(2) what is the impact of cutting the audio after 30 seconds?

- is it really needed? can you not cut the test audio in turns using a provided (or reconstructed) turn-level segmentation?
- what happens with reference words that span 30.0 seconds? are they in or out?
- what happens with semantic methods (SentBERT, SemDist) when we present them multiple turns together or half a turn? this would affect results and could explain why semantic methods don't clearly outperform other utility functions?

**Requested Changes:**

See my comments above.

(1 - required) use standard test sets and show that the authors baseline decoding matches the results that other groups are reporting under identical conditions. or explain why simply using the first 1000 test examples is advantageous

(2) analyze the impact of cutting audio at 30 seconds
- (required) describe their process in more detail: how were the references processed, how many utterances were affected in each corpus
- (recommended) use the full standard test set and/ or experiment with segmented data, especially for the "semantic" methods
  - if only 5 of 1000 utterances are longer than 30 seconds, the effect of cutting may be negligible, but we don't know

(3 - recommended) replace the "reranking" discussion with an analysis of "consensus" decoding as another way of directly optimizing for word error rate (rather than MAP), or drop it entirely
  - I find the "reranking" section is only vaguely connected to the rest of the paper and not particularly insightful. if the goal is to compare MBR with other methods that achieve the same goal, I would recommend comparing to "consensus" decoding

(4 - recommended) add a discussion of pruning to the work. pruning greatly impacts the diversity of hypotheses, especially when used together with different segmentations. what do the various parameters do? is beam applied to the number of hypotheses or the score difference between the best and the hypothesis? etc

(5 - recommended) provide some analysis on why MBR underperforms for short utterances?
- does it "collapse" the hypotheses to some generic and "central" (but uninformative) text?
- maybe break this down in even smaller length groups?

---

> ### Author Response · Authors · 2026-03-07
> **Thank you very much for the comment**
>
> We thank the reviewer for the thoughtful feedback and the recognition of decoding as an under-appreciated topic. We appreciate the opportunity to clarify our experimental design and improve the rigor of our analysis.
>
> > (1 - required) Use standard test sets... or explain why simply using the first 1000 test examples is advantageous.
> > (2 - required) Analyze the impact of cutting audio at 30 seconds... describe the process in more detail.
>
> We apologize for the confusion regarding our data selection process. To be clear: we did not cut or truncate any audio files. We excluded samples longer than 30 seconds from the evaluation.
> We chose this approach to scientifically isolate the performance of MBR decoding from the effects of "long-form" heuristics. The Whisper model has a hard input limit of 30 seconds. To process longer audio, one must use a "stitching" algorithm (chunking audio, transcribing independently, and merging text). This introduces complex hyperparameters (overlap, merge strategies) that are distinct from the decoding strategy itself. By focusing on samples <30s, we ensure that the improvements observed are strictly due to MBR decoding and not due to interaction with a stitching algorithm.
> However, to address your concern regarding standard benchmarks, we have now run the full LibriSpeech Clean test set (all 2,620 samples). We use the sequential long-form algorithm following the recommendation in the README of the whisper-large-v3 model (https://huggingface.co/openai/whisper-large-v3/blob/main/README.md).
>
> Impact of exclusion: In this dataset, only 9 samples (approx. 0.3%) exceeded 30 seconds. Their exclusion has a negligible impact on the overall metrics.
>
> Results: As shown below, the trend remains consistent: MBR decoding continues to outperform Beam Search on the full standard test set.
>
> | model | whisper-large-v3 | | |
> | :--- | :--- | :--- | :--- |
> | **Metric** | **wer** | **metricx** | **sentbert** |
> | Beam 1 | 0.036 | 1.744 | 0.064 |
> | Beam 5 | 0.037 | 1.731 | 0.064 |
> | Beam 20 | 0.036 | 1.743 | 0.062 |
> | MBR 64 | 0.029 | 1.631 | 0.053 |
>
> > (3 - recommended) Replace the "reranking" discussion with an analysis of "consensus" decoding
>
> We agree that the connection between MBR and consensus decoding is significant. MBR decoding can be viewed as a generalized form of consensus decoding: it selects the hypothesis that minimizes risk with respect to the distribution of other hypotheses, effectively finding the "centroid" or consensus of the N-best list. We will revise the text to explicitly discuss MBR within the framework of consensus decoding, highlighting that MBR with token-level error metrics (like WER/Levenshtein) functions similarly to traditional ROVER-style consensus.
>
> > (4 - recommended) add a discussion of pruning to the work. pruning greatly impacts the diversity of hypotheses, especially when used together with different segmentations. what do the various parameters do? is beam applied to the number of hypotheses or the score difference between the best and the hypothesis? Etc
>
> We used the standard beam search implementation provided in the HuggingFace transformers library. In this setting, "pruning" is defined strictly by the beam width (number of active hypotheses maintained at each step). We did not apply additional heuristic pruning strategies, such as threshold pruning (discarding hypotheses based on a score difference from the best path) or diversity penalties. We will add a "Decoding Configuration" subsection to explicitly detail these parameters to ensure reproducibility.
>
> > (5 - recommended) Provide some analysis on why MBR underperforms for short utterances?
>
> Thank you for this interesting question. We analyzed the failure cases in the AMI-IHM corpus (meeting recordings) and found that short utterances in this domain are predominantly backchannels or non-lexical fillers (e.g., "hmm", "yeah", "mm").
> The Metric Mismatch: The utility function used for MBR (e.g., BLEU) often yields zero overlap for short sequences if a single token is missed or substituted (e.g., "yeah" vs "yes"), leading to a flat utility landscape.
>
> Risk Minimization: In these high-uncertainty, low-information cases, MBR tends to default to the most frequent generic token in the distribution, which may not match the specific filler in the reference. We will add a qualitative analysis of these short-utterance errors in the revision.
>
> Please let us know if there are any other points we can clarify.

---

### Review · Reviewer_RkQs · 2026-03-05

**Summary Of Contributions:**

This paper studies Minimum Bayes Risk (MBR) decoding for automatic speech recognition and speech translation tasks, as previous studies with modern architectures mostly examined text-to-text tasks. On English and Japanese data, MBR generally outperforms beam search, although it is expectedly slower. In particular, the paper examines the impact of the number of samples, sampling algorithm, noise level, speech length and utility function.

**Audience:**

Yes

**Audience Explanation:**

While results are not really unexpected, the paper presents a detailed study of Minimum Bayes Risk decoding for speech-to-text tasks. MBR has not previously been evaluated in this setting as thoroughly as for text-to-text generation. The study systematically examines the impact of different variables, potentially helping practitioners find suitable settings faster for tasks where accuracy takes priority over latency.

**Claims And Evidence:**

Yes

**Claims Explanation:**

The paper clearly shows that MBR can be superior to beam search on speech-to-text tasks (automated speech recognition, speech translation). This is supported by evaluation on multiple languages and datasets. The paper systematically modifies different control variables to examine their impact. The authors further share code to replicate their results.

However, I have some reservations. Using other models outside the Whisper family would strengthen the claims. Some beam search results are suspicious, with identical results across beam widths. Regarding the statement "Intuitively, MBR decoding selects the hypothesis that lies at the center of the sampled hypotheses," I would note that if the sampling process is biased (e.g., toward shorter sentences or overly frequent tokens), the center of the hypotheses may be far from the optimal output. I am also unsure about the statement that beam search is O(GB) (cost to generate hypothesis times beam width). Does this assume that the branching factor (number of hypotheses for each example in the beam before pruning) is constant? Finally, I was slightly confused by "u(y,y′) is a utility function that measures the quality of hypothesis y against reference y′." The y′ refers to other hypotheses, not necessarily a reference output.

**Requested Changes:**

[Critical] Across pages 3-4, update incomplete sentence "This result is consistent with empirical findings showing that larger sample sizes lead to (Freitag et al., 2023)."

[Critical] Verify beam search results. They are identical across different beam widths (B=1, 5, 20) in Table 1 (all models), Table 8 (all languages), and some datasets in Table 2 (VoxPopuli, AMI-IHM), while other datasets show variation. This is unexpected.

[Would strengthen] The statement "Intuitively, MBR decoding selects the hypothesis that lies at the center of the sampled hypotheses" needs qualification. While Section 2.2 discusses sampling methods, it doesn't address what happens when the model's distribution itself is
systematically biased. If the model tends to generate biased outputs (e.g., overly short sentences), even "unbiased" sampling will produce a biased center. Please clarify this limitation.

[Would strengthen] Clarify the claim that beam search is O(GB). Does this assume constant branching factor? Please explain how vocabulary size and pruning operations factor into this complexity.

[Would strengthen] Clarify the description of Equation 3. The phrase "u(y,y′) is a utility function that measures the quality of hypothesis y against reference y′" is misleading, as y′ refers to other sampled hypotheses, not a reference output.

---

> ### Author Response · Authors · 2026-03-07
> **Thank you very much for the comment**
>
> We thank the reviewer for the detailed assessment and the constructive feedback, which has helped us improve the clarity and accuracy of our paper. Below, we address the requested changes and questions.
>
> > [Critical] Across pages 3-4, update incomplete sentence "This result is consistent with empirical findings showing that larger sample sizes lead to (Freitag et al., 2023)."
>
> Thank you for catching this error. We have corrected the sentence as follows:
> "This result is consistent with empirical findings showing that larger sample sizes lead to higher generation quality (Freitag et al., 2023)."
>
> > [Critical] Verify beam search results. They are identical across different beam widths (B=1, 5, 20) in Table 1 (all models), Table 8 (all languages), and some datasets in Table 2 (VoxPopuli, AMI-IHM), while other datasets show variation. This is unexpected.
>
> We have rigorously double-checked our experimental logs and confirmed that the reported results are correct. We found that for these specific models and datasets, the generated transcripts were indeed identical across the tested beam widths (B=1,5,20).
> We attribute this to the highly peaked probability distribution of the Whisper model family. As noted by Radford et al. (2022), performance on benchmarks like LibriSpeech Clean is already near saturation (comparable to human transcription). This suggests the model is extremely confident in its predictions; consequently, the greedy path (top-1) consistently aligns with the optimal path found by wider beams. We will add a note in the experimental section clarifying that this behavior stems from the high confidence of the underlying model.
> > [Would strengthen] The statement "Intuitively, MBR decoding selects the hypothesis that lies at the center of the sampled hypotheses" needs qualification... If the model tends to generate biased outputs... even "unbiased" sampling will produce a biased center. Please clarify this limitation.
>
> We agree with this assessment. Since MBR seeks the mode of the distribution defined by the samples, it naturally reflects any systematic biases present in the underlying model (such as a bias toward shorter sentences). We will amend the description in Section 2.2 to explicitly qualify that MBR centers on the model’s distribution, and thus will preserve model-intrinsic biases.
>
> > [Would strengthen] Clarify the claim that beam search is O(GB). Does this assume constant branching factor? Please explain how vocabulary size and pruning operations factor into this complexity.
>
> Thank you for the question. In our notation, O(GB) assumes that the branching factor (vocabulary size) and pruning operations are architectural constants absorbed into the generation cost term, G. G represents the computational cost of a full decoding step for a single hypothesis. We will clarify in the text that while the vocabulary size impacts the constant factor of G, the complexity scales linearly with the beam width B and the generation cost.
>
> > [Would strengthen] Clarify the description of Equation 3. The phrase "u(y,y′) is a utility function that measures the quality of hypothesis y against reference y′" is misleading, as y′ refers to other sampled hypotheses, not a reference output.
>
> We appreciate the suggestion to improve clarity. We meant that u(y,y′) measures the quality of hypothesis y by treating y′ as if it were the ground truth reference. We will revise the text to explicitly state that y′ serves as a pseudo-reference drawn from the peer hypotheses, rather than a gold-standard reference.
>
>
> Please let us know if there are any other points we can clarify.

---

### Author Response · Authors · 2026-03-13
**Manuscript updated**

We thank the reviewers for your detailed and constructive feedback.
We have revised the manuscript accordingly (https://openreview.net/pdf?id=I6iLWhRIsf).


Please let us know if you have any other concerns to our manuscript.

---

### Author Response · Authors · 2026-04-20
**Status inquiry**

Dear representatives,

I am following up on submission #6969, which has been under evaluation since our last revision update on March 13th.

I would be helpful if you could provide a brief update on the expected timeline for the final decision.
We remain available should any further clarifications be needed.

Sincerely,

Authors

---

> ### Comment · Action_Editor_Xibi · 2026-04-20
> **All recommendations received**
>
> One reviewer was late submitting their official recommendation due to a job change and other conference reviewing duties. I have now received all three recommendations and I will be able to make a decision this week.
>
> AE

---

> > ### Author Response · Authors · 2026-04-21
> >
> > Thank you for the update.
> > We appreciate for letting us know, and we look forward to receiving your decision.
> >
> > Authors

---

### Author Response · Authors · 2026-05-09
**Submitted the camera ready**

Thank you very much for your work for coordinating the review process.

I have now submitted the camera-ready manuscript.
The content remains largely unchanged, with the primary update being the deanonymization of the authors and affiliations.

Please let me know if there is anything further required on my end.

---

### Decision · Action_Editor_Xibi · 2026-04-23

**Recommendation:** Accept as is

**Audience:**

Yes

**Audience Explanation:**

Decoding algorithms matter, but they have not received much attention recently. This paper is a good addition to the conversation.

**Claims And Evidence:**

Yes

**Claims Explanation:**

The most significant flaw in the original manuscript, namely that it did not report results on any full, standard test set (as noted by Reviewer BC3i) has been corrected in the revision, which now provides full results on Librispeech.

More broadly, the revised manuscript has a clearer discussion of MBR decoding, results on non-Whisper ASR models, and deeper analysis of experimental results.

The main message of the paper, that MBR decoding should be considered for ASR and AST tasks where accuracy is to be prioritized over latency, is well supported by a set of experimental results on different tasks, noise levels, languages, and models.

---

> ### Author Response · Authors · 2026-04-26
>
> Dear Action Editor and Reviewers,
>
> Thank you for your time in organizing and reviewing our paper.
> We will submit the camera-ready version once it's ready.
> Once again, thank you for your constructive comments throughout the review process.
>
> Best regards,
>
> Authors